# Senescence of Tumor Cells in Anticancer Therapy—Beneficial and Detrimental Effects

**DOI:** 10.3390/ijms231911082

**Published:** 2022-09-21

**Authors:** Wiktoria Monika Piskorz, Marzanna Cechowska-Pasko

**Affiliations:** Department of Pharmaceutical Biochemistry, Medical University of Bialystok, Mickiewicza 2A, 15-222 Białystok, Poland

**Keywords:** senescence, cancer, therapy, senolysis, senolytics, senostatics, SASP, prosenescence therapy

## Abstract

Cellular senescence process results in stable cell cycle arrest, which prevents cell proliferation. It can be induced by a variety of stimuli including metabolic stress, DNA damage, telomeres shortening, and oncogenes activation. Senescence is generally considered as a process of tumor suppression, both by preventing cancer cells proliferation and inhibiting cancer progression. It can also be a key effector mechanism for many types of anticancer therapies such as chemotherapy and radiotherapy, both directly and through bioactive molecules released by senescent cells that can stimulate an immune response. Senescence is characterized by a senescence-associated secretory phenotype (SASP) that can have both beneficial and detrimental impact on cancer progression. Despite the negatives, attempts are still being made to use senescence to fight cancer, especially when it comes to senolytics. There is a possibility that a combination of prosenescence therapy—which targets tumor cells and causes their senescence—with senotherapy—which targets senescent cells, can be promising in cancer treatment. This review provides information on cellular senescence, its connection with carcinogenesis and therapeutic possibilities linked to this process.

## 1. Introduction

Cellular senescence is a process resulting in stable cell cycle arrest, which restricts ability of cells to proliferate [1]. It is considered that this state might be a response to chemotherapy, both genotoxic and oxidative stress, oncogenic activation, shortening of telomeres, irradiation or mitochondrial disorders [2]. As a result of senescence, higher lysosomal activity, rearrangement of chromatin, metabolic deregulation, apoptotic stimuli resistance, DNA damage and raised cytokine secretion occur [3,4]. What is more, senescent cells secrete a variety of factors, constituting the senescence-associated secretory phenotype (SASP), which have a tremendous impact on the whole cellular environment [5].

Depending on the duration of exposure and intensity of the senescence inducer, acute and chronic senescence can be distinguished [6]. The first one is a rapid process usually triggered in response to inflammation, embryogenesis or wound healing [7]. Chronic senescence is a state caused by feedback loops and occurs when the exposure to the senescence factor is prolonged [1] or as a result of sustained unscheduled cell damage [4].

Although it might seem that senescence mainly affects the aged organism, senescent cells are present at every stage of life (although their number increases with age) and play a significant role in the functioning of the body [8]. It is believed that senescence is one of the major mechanisms of antitumor defense. What is more, it is also involved in the wound healing process, and during embryogenesis—in tissue formation [3]. However, there are reports of negative effects that senescence has on the body. Through SASP, senescent cells release pro-inflammatory cytokines and chemokines, disrupting stem cells and tissue regeneration and disturbing homeostasis [9]. Due to the fact that the number of senescent cells increases with age, there is a possibility that this condition contributes to the occurrence of age-related diseases, including Alzheimer’s disease and cancer progression. There is a chance that pharmacological removal of senescent cells with senolytics will give positive results in the treatment of these maladies [10,11,12]. The connections between senescence and both tumor progression and regression are visible. Combining prosenescence therapies with senotherapies can be promising in cancer treatment. 

## 2. Senescence Induction

Induction of senescence is influenced by various factors, such as DNA damage, telomere shortening, hypoxia, nutrient deficiency, cell stress, oncogene activation, mitochondrial dysfunctions [3]. There is also a possibility of stimulating senescence through cellular metabolism, regulation of apoptosis, response to unfolded protein (UPR), and DNA damage response (DDR) [13].

Initiation of senescence is associated with the cell cycle, when a cell from the G1 phase, instead of going into the synthesis (S) phase, goes into the G0 phase, resulting in cell division inhibition. Activation of this checkpoint usually depends on DNA damage and telomere shortening [6]. Maintenance of this state, which is extremely important in anticancer therapy, is influenced by the cyclin-dependent kinase (CDK) inhibitors—P16^INK4A^ (CDKN2A) and P21^WAF1/CIP1^ (CDKN1A) proteins—regulated by both the suppressor protein P53 and retinoblastoma proteins (RB) (Figure 1) [14,15]. RB1 protein, as well as P107 (RBL1) and P130 (RBL2), are phosphorylated by CDKs. The phosphorylation weakens their ability to repress E2F transcription factors, crucial to cell cycle progression [3]. In senescent cells, RB proteins are constantly activated due to the accumulation of the CDK4/6 inhibitor P16 and the CDK2 inhibitor P21. E2F transactivation is inhibited and consequently, cell cycle arrest occurs. This state is maintained by the SASP, heterochromatinization of E2F target genes and reactive oxygen species (ROS) and cannot be reversed by P53 or RB inactivation [3,16].

### 2.1. Replicative Senescence

According to the Hayflick theory, cells’ ability to replicate is limited [17]. After reaching the limit, they become senescent cells—alive and metabolically active, but unable to divide. The main factor causing replicative senescence is the shortening of telomeres and the lack of telomerase responsible for their extension [18,19]. Telomeres are located at the ends of the chromosomes and are tandem repeats of TTAGGG nucleotides, stabilized by the Shelterin protein complex [18,19]. The majority of cells (except for stem and cancer cells) do not express telomerase responsible for maintaining telomere length [3]. With each replication, the length of telomeres decreases [6]. Eventually, the free end of the chromosome is exposed, which is perceived as a double-strand DNA break (DSB). DSBs activate the DDR [4].

### 2.2. Stress–Induced Senescence

As mentioned before, senescence is associated with DDR. DNA damage caused by the above-described telomere shortening, induction of an oncogene, as well as damage due to the oxidative stress, external factors, and chemotherapeutic agents result in a cellular response [6]. A strong inducer of DDR is also the sugar-phosphate DNA backbone damage—caused by, among others, ionizing radiation (IR) or topoisomerase inhibitors—which may cause DSBs [20]. In case of constant DNA damage signaling, DSB contributes to the enhanced secretion of pro-inflammatory cytokines, such as interleukin-6 (IL-6) and onset of inflammation [21]. Diminished selective autophagy of GATA binding protein 4 (GATA4)—transcriptional factor—also contributes to the induction of DDR-dependent senescence and inflammation. During senescence—due to the action of ataxia telangiectasia mutated kinase (ATM) and ataxia telangiectasia and Rad3-related kinase (ATR)—activation of P16^INK4A^ and P53 pathways occurs. As a result, p62-dependent autophagic GATA4 degradation is inhibited. Activation of nuclear factor kappa B (NF-κB) occurs, which leads to the SASP initiation [22]. This process seems to be independent of P16^INK4A^ and P53 [20]. Furthermore, telomere related factors such as protection of telomeres 1 (POT1) and telomeric repeat-binding factor 2 (TRF2) may impair the activity of DDR via inhibiting ATM and ATR kinases involved in the cell response to DNA damage [23,24]. Eradication of TRF2 or POT1 from telomeric DNA leads to DDR [25].

**Figure 1 ijms-23-11082-f001:**
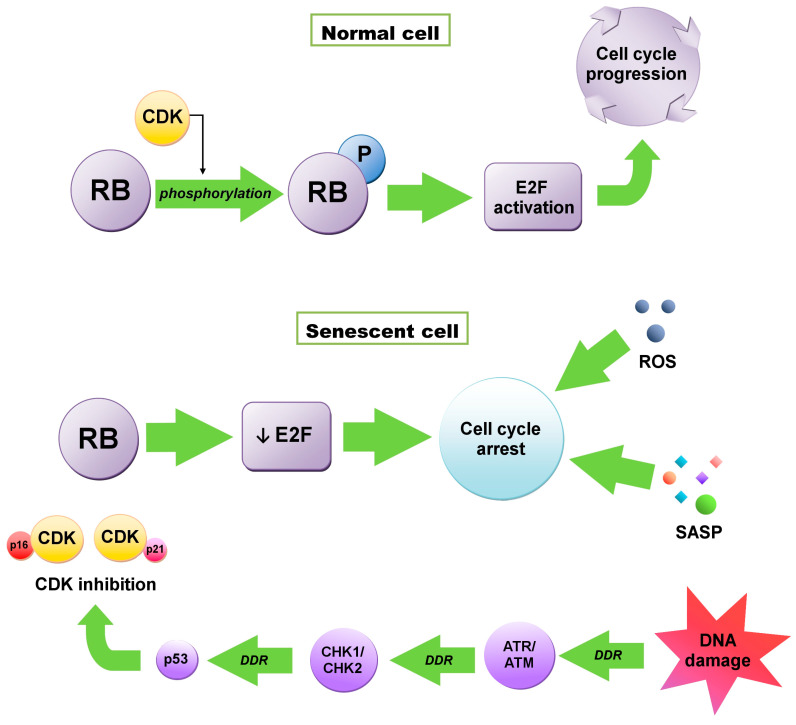
Retinoblastoma (RB)-dependent cell cycle arrest. In normal cells RB is phosphorylated by cyclin-dependent kinases (CDKs). Enhancement of E2F transcription factor activity occurs, which is required for cell cycle progression. In senescent cells, CDKs are inhibited by P16 or P21, which can be activated as a result of DNA damage response (DDR) through ataxia telangiectasia and Rad3-related kinase (ATR)/ataxia telangiectasia mutated kinase (ATM) → checkpoint kinase 1 /checkpoint kinase 2 (CKH1/CHK2) → P53 pathway. E2F repression occurs, which leads to the cell cycle arrest maintained by the senescence-associated secretory phenotype (SASP) and reactive oxygen species (ROS).

Regardless of the inducer, the DDR mechanism is based on the activation of the kinase cascade (first ATR and ATM kinases, then checkpoint kinase 1 (CHK1) and kinase 2 (CHK2)). ATM responds to the DNA DSB and activates the DNA repair mechanism via H2A histone family member X (H2AX) phosphorylation—a crucial step of the production of DNA repair factors (Nijmegen breakage syndrome 1 protein (NBS1), mediator of DNA damage checkpoint 1 (MDC1)/nuclear factor with BRCT domains protein 1 (NFBD1), P53 binding protein 1 (53BP1) and checkpoint proteins. Phosphorylated H2AX (γH2AX) also controls CHK1 and CHK2 phosphorylation, which transmits the signal to P53/P51 proteins [4,26]. Eventually, the P53 protein is phosphorylated, and its activation leads to cell cycle arrest [25,27]. It has been proven that inhibition of DDR, and more specifically ATR, ATM, CHK1, CHK2 kinases, leaves an open door for senescent cells to re-enter the cell cycle [25,28].

As a result of DDR activation, the cell may enter a state of apoptosis or senescence. The path it takes depends on the degree of DNA damage and the possibility of its repair. Initiating cellular senescence through DDR activation is a crucial process in chemotherapy and radiotherapy [6]. In cancer, tumor suppressor genes such as *P53* and *BRCA* are mutated, which is perceived as DNA damage. Damaged DNA accumulates and finally leads to DDR. As a result, tumor cells may enter state of apoptosis or senescence through P53/cyclin E/CDK2/RB pathway [6].

#### 2.2.1. Oncogene–Induced Senescence

Senescence may also be induced by oncogenes; this is considered to be a tumor suppressor mechanism and the first barrier preventing intensified cell proliferation [3]. During oncogene-induced senescence (OIS), cell cycle inhibition is usually mediated by the tumor suppressors—ADP ribosylation factors (ARF) and P16^INK4A^, encoded by *CDKN2A* locus [2]. Certain oncogenes, such as *BRAF*, may induce senescence through the activity of mitochondrial pyruvate kinase [6]. Oncogene expression induces hyperproliferation. Consumption of DNA replication resources increases and finally the stalled replication forks occur, leading to DDR followed by senescence [28,29]. Inhibited expression of cancer suppressors triggers cell proliferation arrest, as exemplified by PICS (phosphatase and tensin homolog deleted on chromosome ten (PTEN) loss-induced cellular senescence), which has recently also been proven to be associated with hyperproliferation and the DDR mechanism [30].

#### 2.2.2. Oxidative Stress

Due to the replicative stress, the accumulation of ROS occurs. ROS induce cellular senescence by affecting various SASP components including regulation of P53/P21 and P16^INK4A^/pRB pathways, enhancement of interleukin 1α (IL-1α) and matrix metalloproteinases (MMPs) synthesis, inhibition of sirtuins, decreasing activity of sarco/endoplasmic reticulum Ca^2+^-ATPase or mammalian target of rapamycin (mTOR) pathway regulation [31]. Furthermore, during OIS, ROS are able to imitate the signalling factors that are associated with promitogenic oncogene functions. Furthermore, the oncogene-induced ROS, arising from nicotinamide adenine dinucleotide phosphate (NADPH) oxidases, can lead to senescence by enhancing the initial hyperproliferative phase caused by the accumulation of DNA damage and altered DNA replication [32].

#### 2.2.3. Metabolic Stress

Metabolic stress is also a senescence inducer. Due to the stress factors—such as acidosis and hypoxia—UPR is triggered. UPR leads to senescence by arresting the cell cycle. Due to the high glucose demand and abnormal circulation, metabolic stress commonly occurs in cancers and the UPR [33].

## 3. The SASP

Senescent cells secrete a specific mixture of factors, including MMPs, cytokines, chemokines, growth factors, and angiogenic factors, which enable them to affect nearby cells. The composition of the molecules varies according to both the senescence inducer and the cell type. This secretome is known as the SASP [3,34]. The SASP promotes the spread of senescence and inflammation to surrounding cells in a paracrine way [35] and also triggers immune response to expel senescent cells [36]. Interestingly, senescent cells themselves are immune to the apoptotic and damaging influence of the SASP and they generate even more of the secretome in response [6].

### 3.1. The SASP Composition

The composition of SASP differs depending on the situation, which may be connected with its antagonistic effects. Early research on the SASP revealed the diversity of molecules secreted by senescent cells, among them chemokines and cytokines: C-X-C motif chemokine ligand 1 (CXCL1), C-C motif chemokine ligand 2 (CCL2), IL-6, interleukin 8 (IL-8), prosurvival factors like glial cell line-derived neurotrophic factor (GDNF), insulin-like growth factor-binding proteins (IGFBPs) and growth modulators such as amphiregulin (AREG) [9]. Nowadays the composition of SASP is better known. Research shows that SASP also contains proteases, ceramides, extracellular matrix (ECM) molecules, signaling factors like bradykinins, hemostatic factors and damage-associated molecular patterns (DAMP) [5,37].

Cell type and senescence inducer seem to be major factors influencing the SASP composition [38]. Mitochondrial dysfunction-associated senescence (MiDAS) results in a distinct senescence response where the interleukin 1 (IL-1)-dependent proinflammatory impact does not occur [39]. The latest research shows that some of the SASP components are common in different cells and different types of senescence and may be some kind of core [37]. The composition of SASP depends on the passage of time as well. During OIS, transforming growth factor β (TGF-β) rich early immunosuppressive secretome turns into the SASP due to the fluctuation of neurogenic locus notch homolog protein 1 (NOTCH1) signaling [40]. What is more, expression of type I interferons (IFN) occurs in late OIS and is partly initiated by the reduction of LINE-1 retrotransposable elements [41]. This may partly explain the variability and sometimes contradiction in SASP effects. As can be seen, senescent cells’ secretome is complex and depends on a plethora of different factors. In order to better understand the SASP, further research is needed.

### 3.2. The SASP Beneficial Effects

Due to the differences in the quantity and nature of the molecules secreted by senescent cells, SASP may have both positive and negative effects on the organism. Beneficial and detrimental effects of some of the SASP components are presented in Table 1. Considering positives, senescent fibroblasts’ SASP, through platelet-derived growth factor AA (PDGF-AA) and cellular communication network factor 1 (CCN1), boost the skin regeneration process [42,43]. Moreover, the SASP reinforces tissue stemness and plasticity, prevents excessive tissue fibrosis, and restores homeostasis in liver fibrosis [44,45]. It can also promote the reprogramming of nearby cells if the tissue is damaged [46]. In addition, the SASP is responsible for the recruitment of immune cells, such as natural killer (NK) cells, T helper type 1 (Th1) cells and macrophages, which play a key role in initial preneoplastic cells elimination and prevent further carcinoma expansion [47]. Moreover, it can skew the polarity of macrophages to tumor-inhibiting state (M1), creating an antitumor environment [48]. Some elements of SASP, such as IL-6, IL-8, IGFBP7, and plasminogen activator inhibitor (PAI-1), support the senescent cell growth arrest [4].

### 3.3. The Role of the SASP in Tumor-Promoting Environment

On the other hand, SASP-induced inflammation may negatively affect cells and even provoke cancer proliferation [4]. IL-1α, which is present in different types of senescence, influences cells in an autocrine way and generates inflammation. It activates NFκB, and enhances IL-6 and IL-8 secretion [80]. Additionally, IL-6 can promote cancer progression in a paracrine way by affecting the IL-6 receptor. There is activation of signal transducer and activator of transcription (STAT3), which leads to transcription of oncogenes and growth regulators, such as c-FOS, cyclin D1, c-MYC, mTOR and mammalian target of rapamycin complex 1 (mTORC1) [81]. It has been proven that as a result of the elimination of senescent cells, the level of proinflammatory cytokines, such as IL-1α, IL-6, and tumor necrosis factor α (TNFα), decreases [82,83], which prevents cancer relapse [84].

In hepatocellular carcinoma, CCL2 was observed to recruit immunosuppressive myeloid cells which restricted NK cells activity leading to further tumor progression [85]. CXCL1 is strongly expressed in both senescent cells and ovarian cancer samples [86]. Secretion of this chemokine can affect stromal cells in a paracrine way and promote tumor invasion and progression [87]. Many of the SASP factors (IL-1α, IL-6, IL-8, C-X-C motif chemokine ligand 12 (CXCL12), CCL2) are also common after anticancer therapy [88] and contribute to physical function decrease, fatigue, loss of appetite and cardiovascular morbidity [89,90,91]. Taking into consideration that senescence is an inherent element of cancer treatment (deliberately or not), it is believed that senescent cells may enhance some of the short-term and long-term side effects of anticancer therapy [87].

As mentioned before, elements of the SASP include growth factors, such as proangiogenic vascular endothelial growth factor (VEGF), which lead to cancer vascularization [92]. Furthermore, MMPs released by senescent cells promote cancer growth [93] and stimulate VEGF-dependent vascularization [94]. Moreover, the SASP is able to induce epithelial-to-mesenchymal transition (EMT), which promotes cancer vascularization and, consequently, tumor growth [34]. Research shows that senescent fibroblasts secretome may support tumor vascularization in xenograft transplants and enhance metastasis and proliferation of premalignant epithelial cells [92,95]. In co-culture composed of senescent lung fibroblasts (line WI38) and preneoplastic epithelial cells, the acceleration of proliferation was observed compared to co-culture with normative fibroblasts [9]. What is more, the enhanced-growth effect was not dependent on senescence inducer and did not occur when co-culturing senescent fibroblasts with normative epithelial cells [9]. Additionally, injection of senescent fibroblasts to breast cancer cells resulted in tumor growth acceleration [96]. What is more, the osteopontin (OPN) secretion increase was detected in senescent stromal cells of murine skin papilloma. Co-injection of OPN-deficient senescent cells was performed and the limitation of cancer growth was observed [97].

Milanovic et al. [98] conducted research to check the influence of chemotherapy-induced senescence on stem-cell-related properties in cancer. In senescent cells, enhancement of an adult tissue stem-cell signature, distinct stem-cell markers and Wnt signaling—a crucial factor in stem-cell restoration—have been observed. Furthermore, treating cells which escaped senescence and never senescent cells with a precise dose of chemotherapy resulted in accelerated and Wnt-dependent growth in previously senescent cells [98]. Wnt signaling is activated in therapy-induced senescence (TIS) and may contribute to higher cancer-initiating potential [98,99,100]. Additionally, in p53-regulatable models of acute leukemia, enhancement of senescence resulted in reprogramming of non-stem bulk leukemia cells to self-renewing stem cells that initiate leukemia [98]. On the other hand, recent studies on skin cancer cells indicated that senescent cells, in fact, promote tumor growth by, among others, P38MAPK and MAPK/extracellular signal-regulated kinase (ERK) up-regulation but do not induce the cancer development [101].

### 3.4. The SASP Regulation

The SASP regulation occurs at several levels, including secretion, mRNA stability, transcription and translation. Paracrine and autocrine feedbacks, which intensify secretion, are also important factors [4]. In control of the SASP regulation are, among others, cyclic GMP-AMP synthase (cGAS)/stimulator of interferon genes (STING) [86,96], DDR [9,21], or P38 mitogen-activated protein kinases (MAPK) [102]. All of them lead to NF-κB and CCAAT/enhancer-binding protein-β (C/EBPβ) activation, which directly controls the transcription of IL-6 and IL-8. These molecules in an autocrine way—through a feed-forward loop—increase the activity of NF-κB and C/EBPβ and intensify SASP [4]. Suppression of C/EBPβ transcriptional activity leads to a decrease in the SASP inflammatory activity [103]. Elimination of NF-κB and C/EBPβ results in IL-8 reduction, which affects OIS growth arrest [104,105]. As can be seen, there are visible connections between SASP, NF-κB activation, DDR and transcription factor GATA4 stabilization.

#### 3.4.1. Epigenetic Regulation

The SASP can be also affected in an epigenetic way. Repression of transcriptional activator and oncoprotein MLL1 causes reduction of proliferative cell cycle regulators and inhibits SASP expression [106]. What is more, recruitment of bromodomain-containing protein 4 (BRD4) to superenhancers activated by senescence seems to be essential for the SASP induction [107]. Chromatic alterations, such as higher expression of macroH2A1 histone [108], reduced H3K9 histone dimethylation [109] or accumulation of H2AJ histone [110] also contribute to SASP regulation.

SASP induction may be provoked by inflammasomes, in particular inflammasome NLRP3, triggered by DAMP, which activates and processes IL-1β [35]. In OIS, inflammasome activation is caused by priming the toll-like receptor 2 (TLR2) by serum amyloids A1 and A2 [111]. A key role in the SASP induction is played by cytosolic DNA, derived from retrotransposable molecules (for instance LINE-1), cytosolic chromatin fragments (CCF) and mitochondrial DNA [3]. DDR and dysfunctional mitochondria promote formation of CCF and trigger the SASP [112]. The cGAS detects cytosolic DNA and synthesize cyclic GMP-AMP (cGAMP), which leads to STING activation [86,96,113]. STING activation results in TANK-binding kinase 1 (TBK1) secretion and IRF2 and NFκB activation, which stimulate IFN-I response [5]. Activation of P38MAPK stimulates activation of NF-κB as well [102].

MiDAS SASP is heavily influenced by metabolic abnormalities in NAD^+^ (nicotinamide adenine dinucleotide) to NADH ratios, which contribute to activation of P53 via influencing 5′AMP-activated protein kinase (AMPK) and enhance SASP [39]. Additionally, increased NAD^+^ salvage caused by a higher level of nicotinamide phosphoribosyltransferase (NAMPT) accelerates mitochondrial respiration and glycolysis in OIS and enhances SASP [114]. In the long run, all of these pathways lead to the induction of SASP.

#### 3.4.2. Post-Transcriptional Control

A major factor in SASP regulation is the mTOR pathway. It has been proven that phosphorylation of 4EBP—translation repressor protein—mediated by mTOR, influences the translation of mitogen-activated protein kinase-activated protein kinase 2 (MAPKAPK2) and IL-1α [80,115]. The stress-activated serine/threonine-protein kinase MAPKAPK2 activity leads to the inhibition of the mRNA-binding protein ZFP36L1, which directs the SASP components to degrade mRNA. The mTOR pathway can regulate the senescent secretome by influencing the stability of SASP mRNAs [115]. Furthermore, in OIS the TOR-autophagy spatial coupling compartment (TASCC) boosts autophagy and protein synthesis which result in the accumulation of SASP molecules [116]. What is more, the research indicates also that polypyrimidine tract-binding protein 1 (PTB1) can control senescent secretome via influencing an alternative splicing of genes taking part in intracellular trafficking (like EXOC7) [117].

#### 3.4.3. DDR and the SASP

In order to induce cytokine secretion, constant DDR is usually needed [21]. Taking into consideration that deficiency of DDR regulators such as NBS1, CHK2 and ATM restrains cytokine expression caused by genotoxic stress, it can be presumed that these factors have a major impact on the SASP induction [28]. Research shows that ATM contributes to removal of macroH2A1.1 histone variant from SASP genes and so participates in SASP genes expression [108]. Activation of P16 protein may enhance senescence and inhibit cells proliferation, but is not sufficient to initiate SASP. On the other hand, P38MAPK activation is not only sufficient but also necessary to induce senescence and SASP even if DNA damage does not occur [102]. Additionally, P38 enhances the activity of NF-κB and, as a result, induces SASP transcripts expression [28]. Furthermore, P53 inhibition intensify SASP and create a proinflammatory environment, which provoke malignant transformation [21].

As can be observed, the factors related to DDR also influence the SASP, which confirms the thesis, that SASP is a complex phenomenon, dependent on many factors. It seems possible to regulate SASP by manipulating the DDR. It is important to search for more links between DDR and SASP, which may expand SASP regulation possibilities.

## 4. Senescence and Cancer

Senescence is considered to be a process that prevents the progression and invasion of cancer cells and is exploited in anticancer therapies. Despite the ample evidence of beneficial effects of senescence in fighting cancer, there are numerous reports of its protumor activity [95,118,119]. In addition, research indicates the detrimental effects of chronic senescence not only on abnormal cells, but also on normal ones [6]. Both positive and negative effects of senescence are presented in Figure 2. Despite negatives, senescence seems to be an interesting way to fight cancer, especially when it comes to senotherapies—drugs that eliminate senescent cells or suppress their SASP. For this reason, getting acquainted with positive and negative effects that senescence and the SASP may have on cancer cells is crucial.

### 4.1. Tumor Suppression

As mentioned before, senescence plays a key role in tumor suppression by enhancing immune surveillance and decreasing malignant cells proliferation. It has been proven that inactivation of senescence signaling leads to the acceleration of cancer development. A lower level of senescence in invasive cancer and higher level in premalignant lesions were observed, which support the thesis that senescence inhibits malignant progression [120,121].

The SASP consolidates cell cycle arrest via a positive-feedback loop. Inhibition of some SASP elements, such as IL-6, C-X-C motif chemokine receptor 2 (CXCR2), IGFBP7 prevents senescence, and the lack of senescence may promote cancer development [104,105,122]. IL-6 and IL8 influence SASP through CXCR2. DDR activation and enhanced ROS production occur, which enhance cell cycle arrest [104,123]. Furthermore, the SASP affects nearby normative cells in a paracrine way and induces stable cell cycle arrest, which prevents neoplastic transformation [35,124].

#### Immunosurveillance

Senescent cells can, through SASP, activate senescence surveillance—an anticancer immune response in order to inhibit progression of malignant cells [47]. This mechanism is associated with antigen-specific CD4(+) T cells activity and enhances an immune response to cancer tissue. Additionally, SASP contributes to NK cells recruitment and modification of macrophages polarization in order to stop tumorigenesis [48,125].

Moreover, the activity of P53 has major impact on cancer progression [20]. Restoration of its functions induces senescence and tumor suppression in sarcomas, liver carcinomas and lymphomas, and is connected with immune response (immunosurveillance) and chronic inflammation caused by SASP [126,127].

To sum up, senescence is a promising element of anticancer therapies. Stable cell cycle arrest prevents excessive proliferation and, consequently, cancer progression. Furthermore, as a result of the SASP, an enhancement in the immune system functioning occurs, which contributes to the elimination of malignant and premalignant cells.

### 4.2. Tumor Promotion

Besides the positive effects of senescence, there are also negative ones. The majority of anticancer treatments impact the whole organism, which leads to accumulation of senescent cells in the tumor environment [15]. The role of the SASP in creating a tumor-promoting microenvironment was described above (Section 3.3) Moreover, the accumulation may enhance a commencement and progression of some age-related chronic diseases like fibrotic, cardiovascular or neurodegenerative illnesses [128]. 

#### 4.2.1. Invasiveness

Studies show that co-culturing breast cancer cells with senescent ones enhances migration through a porous membrane [9]. IL-6 and IL-8—the SASP components probably have the major impact here. It has been proven that addition of recombinant IL-8 and IL-6 enhances the preneoplastic epithelial cells invasiveness while inhibition of these interleukins decreases it [9]. MMPs also impact cancer cell invasion. Degradation of the extracellular matrix influences the permeability of capillaries, providing growth factor as well as mitogens and supporting the spread of tumors [93,129].

#### 4.2.2. EMT

The SASP influences cancer development by promoting EMT [15]. As a result of supplying non-aggressive breast cancer cells with the medium of senescent fibroblasts, the miscellaneous features of EMT were observed, such as accelerated expression of vimentin and reduction of both E-cadherin and β-catenin [9]. What is more, incubating mesothelioma cells with senescent mesothelioma cells media (the senescence inducer was chemotherapeutic—pemetrex) leads to enhancement of EMT hallmarks, for instance vimentin upregulation [130]. This supports the thesis that EMT may be promoted in non-senescent cells by senescent tumor cells. The crucial factor of EMT genes transcription is STAT3 [15]. As already mentioned, STAT3 is influenced by the activity of IL-6 and IL-8, and studies show that these interleukins contribute to the induction of EMT in cancer cells in vitro [131,132,133].

#### 4.2.3. Immunosenescence

Immune senescence, especially T-cells immunosenescence, is connected with immune system ageing and can contribute to tumor development by influencing immune surveillance [6]. T-cells senescence may be induced by variety of factors, such as uncontrolled inflammation or chronic T-cells stimulation [134,135]. Senescent T-cells are metabolically active and release cytokines, such as TNFα and IL-6 [134]. Furthermore, they are able to suppress the responder T-cells proliferation, which contribute to cancer progression [136]. On the other hand, senescent T-cells also play an anticancer role by influencing the fate of macrophages [137].

It has been reported that co-culturing tumor cells with premalignant senescent hepatocytes results in NK cells dysfunction and immature myeloid cells recruitment, caused by SASP, and leads to cancer progression [85]. An immunosuppressive environment reduces the ability of immune cells to invade and target neoplastic cells [138]. The study showed an age-related lung cancer progression in mice caused by myeloid-derived suppressor cells (MDSCs), the amount of which increases with age [139]. Moreover, the senescent stromal cells in the skin ageing model were sufficient to recruit MDSCs which inhibited the T-cells response and, as a result, promoted cancer development [140].

Age-related immune dysfunctions, senescent cells accumulation and weakened immunosurveillance lead to the establishment of an immunosuppressive environment, which may contribute to worse survivability of oncological patients [85,87]. The greater expression of senescence genes is connected with a shorter relapse-free period and poorer overall survival [85].

The cell cycle arrest that occurs during senescence is usually irreversible. However, under certain circumstances, and mostly in tumor cells, the re-entry into the cell cycle may take place [98,141,142,143]. It is believed that the failure of senescence may occur due to the loss of one of the crucial senescence effectors, like P53 or P16^INK4A^ [6]. As a result, cancer progresses in an uncontrolled manner.

## 5. Anticancer Therapies

Cells’ entrance into senescence seems to be an inevitable part of anticancer therapies. This process may lead to cancer regression via proliferation suppression and activation of SASP-dependent immune response, as well as to cancer relapse and invasion enhancement. The link between particular cancer therapies and induction of senescence should be considered.

### 5.1. Chemotherapy

Chemotherapy can lead to apoptosis or senescence depending on the functionality of tumor suppressors like P16^INK4A^ and P53 or the duration and intensity of stimuli [6]. Many chemotherapeutic agents cause DNA damage and lead to DDR, which, as mentioned before, can cause apoptosis or senescence. Higher doses of chemotherapy usually lead to cell death, whereas moderate doses lead to cell cycle arrest [6]. Due to the fact that senescent cells remain in the body and secrete SASP molecules, there are concerns that these cells may be a potential reservoir for cancer relapse. In addition, senescent cells can gain a “stem cell”-like phenotype and promote tumor development [84,144], but on the other hand, they may reinforce the immune surveillance, induce senescence of surrounding cells and consequently inhibit cancer proliferation.

It has been experimentally established that doxorubicin induces senescence in a mouse breast carcinoma model (MMTV-Wnt1). Increased P21 expression, C-C motif chemokine ligand 5 (CCL5), C-X-C motif chemokine ligand 5 (CXCL5) and eotaxin expression, as well as senescence-associated beta-galactosidase (SA-β-galactosidase) activity were observed. Those SASP factors stimulate the malignant transformation of nearby normative cells [145]. CCL5 increases the invasiveness of cancer cells by enhancing the expression of MMP9 and MMP2 and stimulates their proliferation by upregulating c-myc and cyclin D1 [146]. CXCL5 promotes metastasis by protein kinase B (AKT)/glycogen synthase kinase 3 beta (GSK3β)/β-catenin and VEGF activation [147,148]. Eotaxin contributes to tumor invasion via MMP3 activation [149]. Interestingly, elimination of doxorubicin-induced nonmalignant senescent cells limits tumor growth and relapse [84]. However, the role of senescence in chemotherapy is more complex due to the varied SASP composition, depended on cell type and senescence inducer.

### 5.2. Radiotherapy

Radiotherapy uses high-energy electrically charged ions to affect tumor cells through ROS and cause DNA damage, which triggers DDR [6,15]. There is a link between the dose of IR and induced DNA damage. Major damage leads to apoptosis, whereas minor damage causes senescence [6]. Information on DNA damage is detected by various proteins and transmitted to effectors and transducers of the DDR mechanism. Cell cycle arrest takes place to provide time required for DNA repair, and if this is not possible, apoptosis or senescence occur [150]. Numerous studies demonstrate that senescence in radiotherapy is dependent on P53 status [151,152]. There is evidence that securing protein participation in repair and replication of DNA influences the cell path (apoptosis or senescence) in IR-exposed tumor cells [153]. The presence of securin in colorectal cancer cells, when exposed to IR radiation, leads to an apoptosis, while its absence results in senescence [153]. The fate of glioblastoma cells after IR exposure depends on the status of PTEN. Decreased PTEN levels result in senescence and increased levels cause apoptosis [154]. IR-induced senescence in lung tumor cells is probably regulated by miR-34a [155]. These examples show that more research is needed into the factors influencing the response to radiotherapy.

Importantly, radiotherapy accelerates an immune response which increases the target cells’ immunogenicity [156]. What is more, IR-induced senescence leads to tissue fibrosis, which is a serious complication, especially in lungs [157] and is connected with skin ulceration due to radiotherapy [158]. Senotherapy may help to mitigate this response.

### 5.3. Prosenescence Therapies

Taking into consideration that senescence limits tumor growth, the idea was that enhancing the senescence of cancer cells could be an option in anticancer therapy. In contrast to chemotherapy, which affects both normal and cancer cells, this approach targets only tumor cells [1].

#### 5.3.1. Telomerase Inhibition

Bearing in mind that lack of telomerase leads to telomeres shortening and consequently to replicative senescence, inhibiting telomerase appeared to be an interesting strategy for inducing senescence in cancer cells. Due to the complexity of the telomerase complex, various telomerase inhibition strategies have been developed, such as chemical telomerase inhibitors [159], inhibitors from microbial sources, antisense oligonucleotides, nucleoside and oligonucleotides [160], gene therapy constructs, molecules targeting telomerase RNA [161,162], molecules targeting telomeres and telomerase-associated proteins, and molecules targeting human telomerase reverse transcriptase (TERT) [1]. The first reported telomerase inhibitor was zidovudine (azido-2,3-dideoxythymidine or azidothymidine), which demonstrated some rate of tumor regression both alone and in combination. Better results were observed with imetelstat—an antisense oligonucleotide—which showed effectiveness in vitro against different cancers [163,164,165]. Many other telomerase-based strategies are currently in different phases of clinical trials and show interesting results [166].

#### 5.3.2. Topoisomerase Inhibition

As mentioned previously, inhibitors of topoisomerase I and II, such as etoposide or doxorubicin, are able to induce senescence of cancer cells. The mechanism of action of these drugs contains deregulation of DNA strands religation [167,168]. It has been proven, that doxorubicin cause breaks in distal chromosomal sequences leading to telomere dysfunction [169]. Decreased levels of RB, P107, and increased levels of P130 has been observed [170]. Furthermore, there was an enhancement of the P16 and P21 proteins expression, which confirmed the arrest of the cell cycle [171,172].

#### 5.3.3. Modulation of Cell Cycle Machinery

Cell cycle arrest is an inherent part of a senescence response. During senescence, the higher expression of CDKs inhibitors such as P15, P21^WAF1/CIP1^, P27, P16^INK4A^ occurs [173,174]. Taking this into consideration, there was an idea that an enhancement of CDK inhibitors level might contribute to prosenescence therapy. CDK inhibitors prevent the phosphorylation of RB and lead to cell cycle arrest [175,176], which results in quiescence state. There is evidence that CDK4/6 inhibitors, such as ribociclib and amebaciclib, have the ability to induce the senescence [177,178]. Furthermore, the latest research on palbociclib established its ability to induce reversible senescence in ER+ breast cancer cells lines T47D and MCF7 and complete senescence in line CAMA 1. The state is assumed to be connected with mTORC1 activity, which promotes stable growth arrest in CDK4/6 inhibitor-induced senescence. Genetic depletion of a negative mTORC1 regulator—tuberous sclerosis complex 2 (TSC2)—in cell lines MCF7 and T47D, resulted in stable growth arrest after palbociclib treatment [179]. What is more, inhibition of CDK2 triggers the senescence dependent on the c-myc oncogene in different cells [180]. In conclusion, CDK inhibitors may have therapeutic importance as prosenescence factors.

#### 5.3.4. P53 Targeting

Given the effect that P53 has on the induction of senescence, P53 activating factors seem to have significance as a prosenescence compounds. In wild-type P53 tumors, the mouse double minute—2 homolog (MDM2)/P53 interaction was inhibited, which resulted in increased P53 activity [181]. Deacetylase sirtuin 1 (SIRT1)—involved in P53 regulation via deacetylation of P53—causes degradation and ubiquitination of this protein and suppresses its activity [182,183]. It has been proven that inhibitors of SIRT1 induce senescence in tumor models [184]. In mutant P53 tumors, senescence has been triggered by small molecules restoring wild-type activity, such as MIRA-1, PRIMA-1 [185] and its analog APR-246 [186,187]. In the absence of P53, senescence has been induced by adenoviral P53 vector [188]. There is also evidence that Src family kinase and receptor tyrosine kinase (c-Kit) inhibitors such as dasatinib induce P53-mediated senescence [189].

#### 5.3.5. Oxidative Stress

Oxidative stress and DDR are well known senescence inducers. Research indicates, that CopA3—D-type disulfide dimer peptide derived from coprisin—a defensin-like antimicrobial peptide, is able to induce cellular senescence through oxidative stress. As a result of CopA3 activity, free radicals are released and cause a DNA double-strand breaks, which lead to DDR followed by senescence. It seems possible to induce senescence with other oxidative stress inducers [190].

#### 5.3.6. SASP Reprogramming

SASP has immense influence on the tumor environment and can affect the cancer progression in both positive and negative ways. Reprogramming SASP may be a key factor in anticancer therapies.

Inhibition of janus kinase 2 (JAK2) leads to the SASP reprogramming and reactivates the senescence immune surveillance [191]. Simvastatin, an inhibitor of 3-hydroxy-3-methylglutaryl-coenzyme A reductase (HMG-CoA), downregulates SASP and suppresses the SASP-mediated growth of breast cancer cells [192]. mTOR–SASP regulator inhibition suppresses SASP and its protumorigenic activity [115]. Additionally, it decreases the ability of senescent fibroblasts to enhance tumor growth. However, rapamycin (mTOR inhibitor) disrupts also the senescence surveillance and paracrine senescence—two significant factors in cancer suppression [115], which show once again the duality of SASP effects on tumor growth.

SASP reprogramming may be a promising strategy in anticancer therapies. It is essential to look for factors that mitigate the negative effects of SASP while maintaining suppressive activity. Further search for prosenescence compounds can expand therapeutic possibilities in cancer treatment, especially when it comes to senotherapies.

### 5.4. Elimination of Senescent Cells

#### 5.4.1. Senotherapies

Senotherapeutics are a new group of medicaments, which restrain the senescent cells’ deleterious influence. They can be divided into categories: senolytic drugs—exterminating senescent cells, and senostatic drugs—suppressing the SASP [6]. Research on senotherapies, in particular on senolytics, indicates their therapeutic value, especially as a support to chemotherapy. It is believed, that these drugs may ameliorate treatment resilience and target tumor senescent cells. Examples of manipulating senescence in tumor cells in order to achieve therapeutic benefits are presented in Figure 3. What is more, riddance of senescent cells induced by chemotherapy alleviates many of the side effects such as cardiac functional impairments, decreased activity or fatigue in a mouse model [84]. Using quercetin and dasatinib in a mouse model in order to remove senescent cells resulted in decreased radiotherapy adverse effects, increased cardiac function and extended life expectancy [193]. 

Targeted elimination of senescent cells is a promising strategy in cancer treatment, especially combined with traditional methods [87]. It has been proven that quercetin has an ability to induce senolysis. The exact mechanism has not been recognised yet; however, there are reports that quercetin inhibits the phosphoinositide 3-kinase (PI3K)-AKT pathway and, as a result, disrupts the activity of the antiapoptotic protein BCL-XL [194]. Another natural flavonol with senolytic activity is fisetin, which is considered to be twice as effective as quercetin [195]. Despite the potent effectiveness of these substances in inducing senolysis, the lack of knowledge about their specific mechanism precludes their clinical use [167]. Additionally, the combination of these senolytic therapies with prosenescence therapies failed to demonstrate effectiveness in animal models of liver cancer [196].

**Figure 3 ijms-23-11082-f003:**
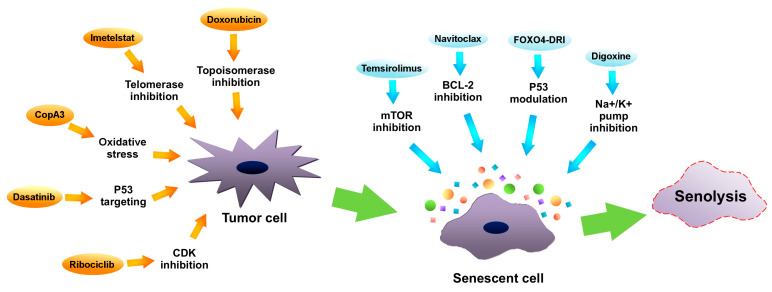
Manipulating senescence in order to eliminate tumor cells by senolysis. Inhibitors of telomerase (such as imetelstat), inhibitors of topoisomerase (i.e., doxorubicin), cyclin-dependent kinases (CDKs) inhibitors (like ribociclib), oxidative stress inducers (such as CopA3) as well as P53 protein modulators (i.e., dasatinib) (orange arrows) may promote tumor cells senescence. Subsequently, via using mammalian target of rapamycin (mTOR) inhibitors (such as temsirolimus), B-cell lymphoma 2 (BCL-2) inhibitors (like navitoclax), Na^+^/K^+^ pump inhibitors (such as digoxine) or P53 modulators (like FOXO4-DRI), senolysis of senescent cancer cells can be induced.

In senescent cells, a high level of BCL-2—antiapoptotic protein—has been observed [197]. The observation led to the idea of using BCL-2 inhibitors as senolytic compounds. It has been proven that navitoclax (ABT-263)—a pan-BCL-2 inhibitor, disturbs antiapoptosis associated with senescence and causes the removal of senescent cells [198]. In addition, it demonstrated an ability to exterminate senescence-like breast (MDA-MB-231) and ovarian (OV1946, OV4453) cancer cells in xenografts and culture, which were previously treated with PARP inhibitor—olaparib [199].

Research shows also that panobinostat—the histone deacetylase (HDAC) inhibitor—initiates removal of senescent cells (senolysis) of head and neck squamous carcinoma (HNSCC) and non-small-cell lung carcinoma (NSCLC) cells, pretreated with chemotherapy [200]. Moreover, research indicates the significant efficiency of cardiac glycosides, especially digoxin, in promoting senescent cancer cells senolysis, pretreated with various chemotherapeutics (doxorubicin, bleomycin, gemcitabine, palbociclib, etoposide) [201]. Cardiac glycosides inactivate the Na^+^/K^+^ ATPase pump and disturb cells’ electrochemical gradient, which leads to acidification and depolarization. Senescent cells are more vulnerable to cardiac glycosides due to their depolarized plasma membrane and increased hydrogen ion concentration [201]. Moreover, cardiac glycosides are also able to induce apoptosis through altering NOXA protein, caused by Na^+^/K^+^ transport inhibition [202]. 

The SASP inhibition (senostasis) may contribute to cancer therapy response improvement. Simultaneous treatment with doxorubicin and metformin (the SASP inhibitor) resulted in growth suppression of breast cancer cell xenografts in mice [203]. In addition, a similar effect was observed after rapamycin treatment in prostate cancer cell xenografts pretreated with mitoxantrone [80].

Inhibition of mTOR—a SASP regulatory factor—leads to senolysis of senescent cancer cells. The combination of the mTOR inhibitor temsirolimus (CCI-779) with chemotherapy (docetaxel) resulted in a strong antitumor activity in breast (MDA-MB-468) and prostate (PC3) xenografts and cancer cells [204]. Protein kinases are enzymes that phosphorylate other proteins in the cell and thus regulate their activity. In normal cells, the activity of protein kinases is tightly regulated, while in cancer cells it often gets out of control and is excessive. This disrupts the functioning of many cell pathways, and as a consequence leads to the intensification of cell division and uncontrolled tumor growth. Inhibition, i.e., inhibiting the excessive activity of protein kinases, is a therapeutic goal of the discussed group of anticancer drugs. Temsirolimus inhibits the activity of the mTOR protein kinase. This kinase controls not only cell division, but also the synthesis of HIF transcription factors, which regulate the tumor’s ability to adapt to oxygen deficiency conditions and produce the factor responsible for the formation of tumor vasculature (VEGF). Inhibition of mTOR kinase activity by temsirolimus limits not only the division of neoplastic cells but also lowers the level of HIF and VEGF proteins in the tumor or its environment, which inhibits the development of tumor vascularization (antiangiogenic effect) [204]. Other research showed that using XL413 in order to inhibit CDC7 (DNA-replication kinase) induced selective senescence in P53 mutated (MHCC97H, Huh7) liver cancer cells. Further addition of mTOR inhibitor (AZD8055) enhanced the tumor suppression [205]. However, disturbing the SASP does not induce senolysis every time. Cell lysis due to mTOR inhibitor is probably associated with senescent cell type [15].

Another way to induce senolysis is by modulating the activity of the P53 protein. The FOXO4-DRI peptide disrupts the P53–FOXO4 interaction, causing P53 nuclear exclusion and subsequent selective apoptosis in senescent non-cancerous cells [83]. Another P53 modulator with a senolytic activity is UBX0101, which interferes with the P53–MDM2 interaction, which results in senolysis. However, the interactions between MDM2 and P53 are not specific and can impact normal cells as well. What is more, senotherapies modulating P53 activity can only be effective in case of TP53-wild-type cancers [167].

#### 5.4.2. Immunotherapy

As described in Section 3.3, senescent cells may create a tumor-promoting microenvironment. Furthermore, they contribute to cognitive dysfunction, neurodegeneration, inflammation [206], and bone diseases such as osteoporosis or osteoarthritis [1]. Removing senescent cells from the body can alleviate the detrimental effects of senescence. Such elimination seems to be possible through immunotherapy [207,208]. Senescent cells possess on their surface different ligands, such as MHC I, MHC II, UL16 binding protein 2 (ULBP2), MHC class I chain-related protein A (MICA) and B (MICB), HLA-E, which are recognized by specific immune cells. For instance, senescent, activated stellate cells generate ligands MICA and ULBP2, which activate natural killer group 2D (NKG2D) receptor on NK cells. Different immune cells exhibit distinct ability to identify and remove specific senescent cells, which makes this strategy selective [209].

### 5.5. Combination of Prosenescence Therapy and Senotherapy

Despite the fact, that combining senescence promotors and senotherapeutics is still at an early stage of research, there are reports confirming the effectiveness of this strategy against cancer cells. In research conducted by Lewińska et al. [210], etoposide-induced senescent breast cancer cells (line MDA-MB-231) and senescent human normal mammary epithelial cells (HMEC) were treated with quercetin derivatives. The activity of one of them (QD3) resulted in senolysis of breast cancer cells, whereas HMEC were not affected. Research on pancreatic ductal adenocarcinoma cells [211] demonstrated effectiveness of a selective fibroblast growth factor receptor 4 (FGFR4) inhibitor—BLU9931—in the induction of senescence. Subsequently, the cells were treated with quercetin, which resulted in the death of the tumor cells. In research on prostate cancer cells [212], senescence was induced either by androgen receptor antagonist—enzalutamide (ENZ)—or by androgen receptor agonists at supraphysiological androgen level (SAL). Afterwards, cells were treated with Bcl-2 family inhibitor—ABT263, heat shock proteins (HSPs) inhibitor—ganetespib or Akt inhibitor—MK2206. While ABT263 demonstrated no senolytic activity, MK2206 induced senolysis in ENZ-treated cells. SAL-pretreated cells remained resistant to MK2206; however, they were vulnerable to ganetespib, which presented senolytic activity. In research conducted on TP53 mutant liver cancer cells [205], senescence was induced by inhibition of cell division cycle 7 (CDC7)-related protein kinase. Subsequently, cells were treated with mTOR suppressors, which caused an apoptosis of CDC7 inhibitor-treated liver cancer cells. Moreover, this combination was also applied to in vivo liver cancer models and resulted in tumor growth inhibition [205]. In research conducted by Galina et al. [213] in an immunocompetent orthotopic mouse model of the aggressive triple negative breast cancer subtype, tumor cell senescence was induced by palbociclib. The senolytic agent was nanoencapsulated navitoclax. The trial resulted in reduction of metastases, inhibition of tumor growth and reduction of systemic toxicity of navitoclax. The efficiency of combining senescence inducers and senolytics depends on variety of factors and demands further research. However, the selectivity and effectiveness of the presented approaches demonstrate the significance of these research areas for future cancer therapy.

## 6. Conclusions

The composition of SASP depends on different agents such as cell type, senescence inducer, and time lapse. However, some molecules are present regardless of the cell type and the senescence inducer. This suggests that SASP has core molecules.The SASP can be regulated by a plethora of factors, and many of them have not been recognized yet. Their discovery could positively influence actual cancer treatment strategies.Senescence can support cancer treatment by arresting the cell cycle and activating immunosurveillance.Senescence may promote tumor progression by enhancing angiogenesis, providing growth factors, and protecting cancer cells from the immune system.Senescence and SASP can promote tumor resistance to therapy and lead to disease relapse.Telomerase inhibition, topoisomerase inhibition, cell cycle modulation, P53 regulation, oxidative stress induction and SASP reprogramming are effective in prosenescence therapy. It is advisable to look for more prosenescence compounds.Senotherapy is an effective method of neutralization and elimination of senescent tumor cells. Searching for more senolytic and senostatic compounds is recommended. Moreover, more research is needed on the currently known senolytics.

In conclusion, senescent cells are a crucial part of antitumor defense. Moreover, they are essential to maintain homeostasis as they participate in processes such as embryogenesis, skin regeneration, and immune response. However, their accumulation can lead to cancer progression, chemoresistance, cognitive dysfunction, inflammation, neurodegeneration and escalation of negative side effects of anticancer therapies. For this reason, excessive senescent cells should be eliminated from the organism. Despite detrimental activity, senescence can still be used in anticancer therapy. Combination of prosenescence compounds and senotherapeutics may prove to be a selective and effective anticancer strategy. Senotherapy not only eliminates senescent tumor cells but also reduces the negative impact that senescent cells and the SASP have on the body. Senescence in cancer is a double-edged sword. On the one hand, it is essential in antitumor defense, but on the other hand, it can support tumor progression. In order to mitigate the consequences, the amount of senescent cells in the organism should be under control.

## Figures and Tables

**Figure 2 ijms-23-11082-f002:**
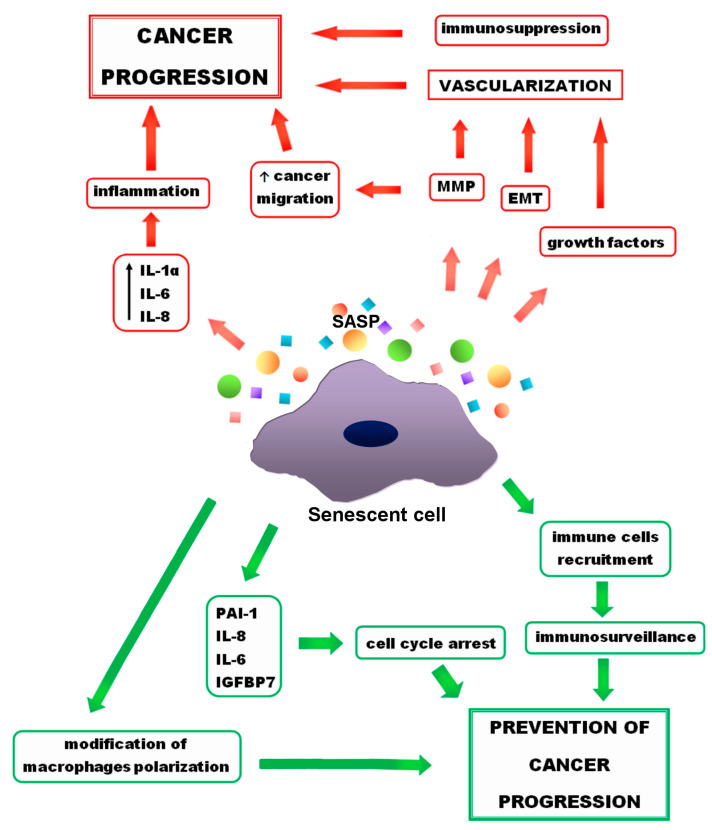
Positive (green arrows) and negative (red arrows) effects of senescence on cancer progression. Growth factors and metalloproteinases (MMP), released by senescent cells, as well as epithelial-to-mesenchymal transformation (EMT) lead to the cancer vascularization. MMPs also support cancer migration. Other (SASP) components, such as interleukin 1 α (IL-1 α), interleukin 6 (IL-6), interleukin 8 (IL-8) result in inflammation, which promotes cancer progression. On the other hand, senescence has also a preventive activity. Some of the SASP components, such as IL-6, IL-8, plasminogen activator inhibitor (PAI-1), insulin-like growth factor-binding protein 7 (IGFBP7) support cell cycle arrest and prevent cancer progression. Moreover, due to the senescence, immune cells recruitment and changes in macrophage polarity occur that prevent cancer progression.

**Table 1 ijms-23-11082-t001:** Beneficial and detrimental effects of the SASP components: CCN1 (cellular communication network factor 1), IL-6, IL-8, IGFBP7 (insulin-like growth factor-binding protein 7).

SASP Component	Beneficial Effects	Detrimental Effects
CCN1	p53 pathway activation → tumor suppression [49]	EMT promotion → cancer progression [50,51]
	Tissue regeneration [52,53]	Cancer cells migration and metastasis [54,55,56,57]
	Wound healing [52,53]	Chemoresistance [58]
IL-6	Epithelial memory of inflammation → tissue regeneration in pancreas [59]	MDSC activation → immunosuppression → lack of response to checkpoint inhibitor therapy [60]
	↑ CCL4, CCL5, CCL17, CXCL10 → T-cells activation [61] Prevention of T-cells apoptosis [62,63] Promotion of T-cells proliferation [64]	Cancer development [59] Inhibition of cancer cells apoptosis [65] Promotion of tumor cells proliferation, angiogenesis and metastasis [65]
IL-8	Response to infection (phagocytosis, oxidative burst, neutrophil extracellular trap [66]	Altering composition of immune cells in tumor microenvironment [66]
	Activation of vascular endothelial cells → cells angiogenesis → ↑ proliferation, migration [67]	Activation of vascular endothelial cells → cancer cells angiogenesis → cancer migration [68]
	Infiltration of immune cells → tumor-reactive cytotoxic mechanism [69]	Infiltration of neutrophils and MDSC → immunosuppression [66,70]
		↑ growth factor secretion by tumor associated macrophages [68]
		Lower survival rate in tumor patients [70]
		EMT induction [66]
IGFBP7	↑CDK inhibitors → cell cycle arrest → tumor suppression [71,72,73]	Infiltration of immune cells in gastric cancer [74]
	↑ E-cadherin, ↓ N-cadherin, ↓ Vimentin → EMT inhibition [75,76,77]	Lower survival rate in gastric cancer patients [74]
	Higher survival rate in melanoma patients [78]	Chemoresistance [79]

## Data Availability

Not applicable.

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
