# Peer review of "Senescence of Tumor Cells in Anticancer Therapy—Beneficial and Detrimental Effects"

_ijms, 2022, doi:10.3390/ijms231911082_

Round 1

Reviewer 1 Report (Previous Reviewer 2)

The authors have addressed my concerns by revising the text and figures and adding a new Table.

Author Response

Reviewer 2 Report (Previous Reviewer 3)

The authors have addressed my concerns raised during the previous rounds of the revieweing process.

Author Response

Reviewer 3 Report (New Reviewer)

The manuscript is well written and organized. I have few suggestions to make which will help to improve the manuscript.

1. A paragraph on induction of a senescence inhibiting migration would be interesting.

2. Senescence is a process to inhibit the cell cycle permanently. However, the presence of these cells for a longer time period is also not healthy. So, I also suggest to dissolve about strategies to get ready of these senescent cells from the body.

3. Discuss the pros and cons of senescence.

4. Figures are very general, I recommend to improve them in terms of information.

5. The below mentioned articles are suitable for citation:

Ogrodnik M, et al.,Aging Cell. 2021. PMID: 33470505.
Dey DK, et al., Mech Ageing Dev. 2021. PMID: 33957217.
Song P, et al., Cells. 2020. PMID: 32164335.

Author Response

This manuscript is a resubmission of an earlier submission. The following is a list of the peer review reports and author responses from that submission.

Round 1

Reviewer 1 Report

The authors have bravely attempted to summarize what has become a large and divergent field. Unfortunately, their broad treatise never really addresses the core question that they have posed – what is the role of senescence in cancer therapy? Instead, the manuscript meanders in a superficial, unfocused manner that is unlikely to be useful in its present form. A principal defect is the over-reliance on previous reviews instead of original research reports. In some instances, other authors’ interpretations of data are misinterpreted. An example of this is in the Introduction where a dubious distinction between acute and chronic senescence is made. The idea that senescence can be “of short duration” directly contradicts the statement that senescence leads to “permanent cell cycle arrest.” Other sections are equally muddled. In particular, the connections among DDR, NAD levels, and SASP responses are alluded to anecdotally without any firm conclusions being drawn. The statement that “senescence is an interesting element of anticancer therapies” is banal and without merit, as is most of the manuscript. The authors need to rethink the goal of the review, and address it more succinctly and conclusively.

Reviewer 2 Report

This is a comprehensive review about the beneficial and detrimental aspects of cellular senescence. The discussion of strategies to harness senescence for cancer therapy is timely and relevant. While the manuscript structure is well organised, I believe the quality of this review could be significantly improved by the following:

1. Significant editing of English language and style (e.g. grammar, punctuation, the use of non-scientific jargon, improper use of definite and indefinite articles).

2. Improving figure quality, for instance, through the use of open-source software. Additionally, figure 2 has an "orphan" arrow not pointing to a specific text box.

3. Inclusion of a table summarising the contexts in which the SASP is beneficial vs. detrimental.

4. Adding an additional figure highlighting possibilities for manipulating senescence.

Reviewer 3 Report

This is a review discussing the role of cellular senescence in cancer progression and cancer therapy concluding that the combination of pro-senescence therapy with senotherapy could be a promising remedy for cancer treatment.

The review contains a lot of information and the respective citations to cover the subject, but I feel it suffers from complexity in terms of structure that makes it very difficult to follow for the reader.

I addition, the authors should decide if they would refer exclusively to the senescence of cancer cells (which should be specified in the tile) or they would also refer to the senescence of stromal cells (only scarce information exists in the original version of the manuscript) [Please also refer to the uploaded file).

The manuscript contains many grammar and syntax errors that need to be corrected (for examples, please refer to the uploaded file).

Proposed structure:

1. - Introduction where cellular senescence is defined and established as an anttumor mechanism. Antagonistic pleiotropy.

- The two types of senescence should be presented, replicative (attributed to telomere shortening) and stress-induced senescence (mostly telomere attrition-independent) with common trait the DDR.

- Types of SIPS could subsequently be presented in separate paragraphs (e.g. OIS, metabolic stress, oxidative stress).

2. - SASP definition and description of its composition

- The importance of SASP (of cancer and stromal cells) in the creation of a tumor-promoting microenvironment

- SASP regulation (original paragraph 3.3)

3.  Senescence and cancer

- Tumor suppression

- Tumor promotion

4. Anti-cancer therapies

- Pro-senescence therapy (paragraphs 6 and 7 should be included, mostly focusing on senescence and not apoptosis)

- Senotherapy

5. Conclusion

Existing information should be organized based on this structure in order to avoid repetitions and to integrate data presented in specified contexts in the proposed paragraphs.

Paragraph 4 (NAD+ level and the senescence) as is seems unsubstantiated. It needs to be included in one pf the above-proposed parts.

Round 2

Reviewer 1 Report

The manuscript has been incrementally improved but is still plagued by poor organization and grammatical issues. Among the more egregious issues:

Line 30: The phrase “cell cycle arrest in diploidal cells, which restricts their capability of proliferation is misleading and redundant. Senescence can occur in cells of any ploidy.

Line 72: Abbreviated terms p16 and p21 are used before introduction on line 77.

Line 139: The phrase the dysfunction of P53 and suppressor genes BRCA, and subsequently, through DDR they enter the state of senescence or apoptosis muddles the role of p53 in senescence. Is it the function or dysfunction of p53 that leads to senescence?

Line 257: The sentence Taking into consideration, that senescence is a significant element of

cancer treatment, it is believed that senescent cells may enhance some of the short-term

and long-term side effects of anticancer therapy is not supported by the preceding statements in the paragraph.

Line 278: The latest research has shown that treating senescent and normal cells with exact dose of chemotherapy results in a higher cancer-initiating potential in senescent cells [20]. Moreover, after repetition of chemotherapy exposure, senescent cells usually maintain the ability to re-enter therapy-induced senescence (TIS), even despite the SASP level reduction - ????

Reviewer 2 Report

The authors have addressed most of my concerns with the exception of Point 3. The authors claim that they did not want to reproduce information present in the figure. My main reason for asking this is that senescence is highly context dependent, so the many of the "detrimental" features can actually be good and not be so generalised. For instance, angiogenesis could actually increase the accessibility of chemotherapeutic drugs or immune cell infiltration to clear tumours. A table showing the features and highlighting the contexts in which senescence is beneficial or detrimental is important and would be informative to readers.

Reviewer 3 Report

This is the revised version of a previously submitted manuscript. The authors have addressed most of my concerns raised during the previous round of the reviewing process, mainly referring to re-organization of the structure of the review.

There are still some grammar and syntax errors that need to be corrected (for examples please refer to the uploaded file, but the whole text needs to be re-cheked).
